# Continual Dialogue State Tracking via Example-Guided Question Answering

**Hyundong Cho[1][*]  Andrea Madotto[2]  Zhaojiang Lin[2]  Khyathi Raghavi Chandu[2]**
**Satwik Kottur[2]  Jing Xu[2]  Jonathan May[1]  Chinnadhurai Sankar[2]**
[1]Information Sciences Institute, University of Southern California, [2]Meta AI
hd.justincho@gmail.com

## Abstract

Dialogue systems are frequently updated to accommodate new services (e.g. booking restaurants, setting alarm clocks, etc.), but naive updates with new data compromises performance on previous services due to catastrophic forgetting. To mitigate this issue, we propose a simple but powerful reformulation for dialogue state tracking (DST), a key component of dialogue systems that estimates the user's goal as a conversation proceeds. We restructure DST to eliminate service-specific structured text and unify data from all services by decomposing each DST sample to a bundle of fine-grained example-guided question answering tasks. Our reformulation encourages a model to learn the general skill of learning from an in-context example to correctly answer a natural language question that corresponds to a slot in a dialogue state. With a retriever trained to find examples that introduce similar updates to dialogue states, we find that our method can significantly boost continual learning performance, even for a model with just 60M parameters. When combined with dialogue-level memory replay, our approach attains state-of-the-art performance on continual learning metrics without relying on any complex regularization or parameter expansion methods.

## 1 Introduction

As conversational digital assistants are becoming increasingly popular and versatile, it is important to continuously update them to accommodate more services.[1] One of their key components is a dialogue state tracking (DST) model that estimates the user's goal, *i.e.* the dialogue state (Williams et al., 2013). The dialogue state is used for queries sent to application programming interfaces to retrieve

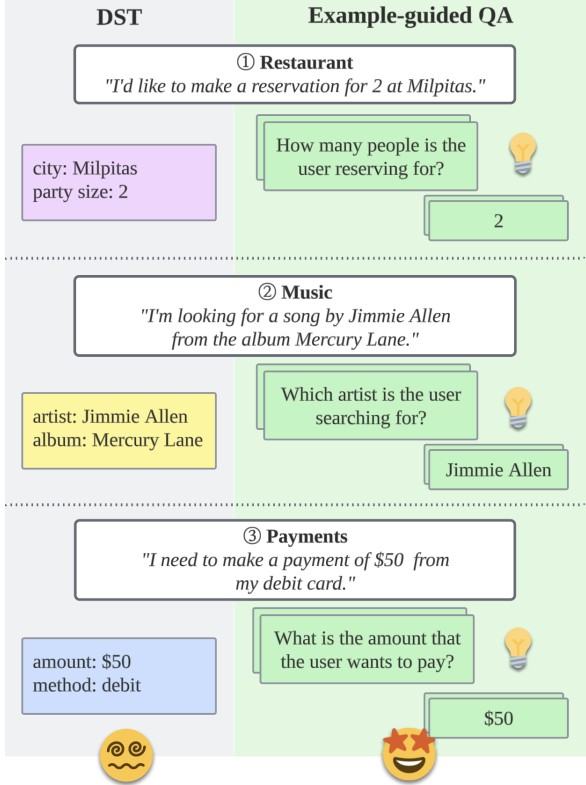

Figure 1: *Left*: When continually learning with the original DST format, DST models need to memorize new slot keys when learning each subsequent service. *Right*: Instead, reformulating DST into a bundle of granular question answering tasks with help from similar examples (symbolized by the light bulbs) makes training data uniform across all services. Learning new services effectively becomes additional training for the general task of example-guided question answering and is more conducive to continual learning.

information that grounds the dialogue model's response.

Unfortunately, naively updating a model for a new service by training with new data causes *catastrophic forgetting* (McCloskey and Cohen, 1989; French, 1999): upon learning from new data, the model's performance for previous services regresses. To mitigate this issue while also avoiding the impracticality of training a model from scratch

---

[*]This work was done while at Meta AI.

[1]In this work, we use *services* and *domains* interchangeably to denote high-level services supported by digital assistants, e.g. setting an alarm or booking a restaurant. *Task* refers to lower-level functions, e.g. question answering, sentiment classification, and dialogue state tracking.

with data from all services each time new data becomes available, three main approaches have been established as generally effective approaches to continual learning (CL): memory replay, regularization, and parameter expansion. Variations and combinations of the three have been applied for DST in previous work (Liu et al., 2021; Madotto et al., 2021; Zhu et al., 2022).

However, most previous work has focused on improving CL performance with service-specific inputs or outputs, a paradigm that limits knowledge transfer between services (left side of Figure 1). This approach introduces a large distribution shift from one service to another since the model needs to memorize service-specific slots that it needs to predict as part of the output. However, DST can become a significantly more consistent task across services by simply reformulating it as a collection of example-guided question answering tasks. Our approach, *Dialogue State Tracking as Example-Guided Question Answering* (DST-EGQA[2]) trains a model to learn to answer natural language questions that correspond to dialogue state slots (right side of Figure 1) with the help of in-context examples instead of predicting service-specific structured outputs all at once without any explicit guidance (left side of Figure 1). We hypothesize that DST-EGQA benefits continual learning because it transforms the DST task to become more granular, easy, and consistent across services.

We discover that this is indeed the case, as our approach leads to significant gains in CL performance without using any of the aforementioned CL approaches or data augmentation methods. Specifically, we transform DST into the TransferQA (Lin et al., 2021) format and add examples from a retriever that is trained to identify turns that result in similar dialogue state updates (Hu et al., 2022). In addition, our approach does not require complex partitioning of the full training set into training samples and retrieval samples. We find that we can use each sample in the training set as both target samples and examples in the retrieval database without causing any label leakage. Also, we experiment with a wide array of retrievers and find that models trained to perform DST-EGQA can be effective even with lower quality retrievers by intentionally training it with subpar examples such that it can learn when to leverage good examples and

ignore bad ones. Lastly, we simply tweak the sampling approach for memory replay to sample at the dialogue-level instead of the turn-level and achieve significant gains to CL performance even with a single dialogue sample, resulting in state-of-the-art performance on the Schema Guided Dialogue (SGD) dataset (Zhu et al., 2022).

In summary, our main contributions are:

1. We show that simply reformulating DST as a fine-grained example-guided question answering task (DST-EGQA) significantly improves continual learning performance by enhancing task consistency across services.

2. We propose a simple but highly effective dialogue-level sampling strategy for choosing memory samples that leads to state-of-the-art performance when combined with DST-EGQA.

3. We share a thorough analysis on DST-EGQA to establish its effectiveness, robustness, and limitations as a method for continual learning.

## 2 Dialogue State Tracking as Example-Guided Question Answering (DST-EGQA)

The goal of continual learning for DST is to sequentially train on a stream of $n$ services $T_1...T_n$ with the goal of minimal degradation, i.e. catastrophic forgetting, of peak performance that was achieved when the model was trained on for each service $T_i$. In this section, we motivate and elaborate on the methodology of DST-EGQA for attaining this goal. Figure 2 presents an illustrated overview.

### 2.1 DST as question answering

Dialogue state tracking (DST) is defined as estimating the beliefs of a user's goals at every turn in a dialogue. It was traditionally formulated as a slot-filling task (Wu et al., 2020; Heck et al., 2020), and more recently as a structured text generation task (Hosseini-Asl et al., 2020; Peng et al., 2021; Su et al., 2022), shown in (0) in Figure 2. If a user were to say *"Find me a 3 star hotel."*, the goal is to deduce `hotel-star = 3`. However, we can also indirectly achieve the same predictions by reformulating DST as a collection of per-slot questions to answer (Gao et al., 2019; Lin et al., 2021). Given the same user request, we can ask our model to answer *"What is the hotel star rating the*

---
[2]Code available at https://github.com/facebookresearch/DST-EGQA

**(0) Domain-specific Structured Text Generation**

Per dialogue turn

| Target Dialogue | → | T5 | → | Service Dialogue State |

**(1) TransferQA: Domain-specific Granular Question Answering**

Per slot per dialogue turn

| Slot Question | Target Dialogue | → | T5 | → | Answer |

**(2) DST-EGQA: Domain-agnostic Example-Guided Question Answering**

Per slot per dialogue turn

| Slot Question | Example Dialogue | Example Slot Value | Target Dialogue | → | T5 | → | Answer |

Retrieve ← → Query

DB (=Training data)

**Sample**

| Target | *I need to stay in a hotel for a meeting. Please find a three star hotel.* |
|---|---|
| Service DS | *hotel-star=3* |
| Question | *What is the hotel star rating the user wants?* |
| Answer | *3* |
| Example | *I need a 5 star hotel please.* |
| Answer | *5* |

Figure 2: DST-EGQA overview. We factor (0) the original dialogue state tracking task into a (1) granular question answering task with the TransferQA format (Lin et al., 2021) and extend it to (2) pair each question with retrieved examples that are provided in-context such that the domain-shift is reduced further to an example-guided question answering task. In TransferQA, the original dialogue state is mapped to templated questions that correspond to each slot key and value pair, which in aggregate request the equivalent information. DST-EGQA applies TransferQA for continual learning and uses the target dialogue as the query to retrieve similar examples from the database, which is formed from the training set excluding the target.

*user wants?"* and have it predict 3. We hypothesize that this question answering approach is more conducive to continual learning because it leverages a general skill that is understandable through natural language. We only need to ask different questions to predict slot values we are interested in. On the other hand, directly predicting a structured dialogue state requires training the model to generate slots that is has never generated before.

To transform DST into question answering as shown in (1) in Figure 2, we leverage the TransferQA (Lin et al., 2021) format. Given $DS_t$, the dialogue state of a dialogue until turn $t$ expressed as (key, value) pairs $\{(s_{t,i}, v_{t,i}) \mid i \in I\}$ for slot $i$, $I = \{1, ..., N_T\}$, where $N_T$ is the number of slots of interest for domain $T$, each $s_{t,i}$ is transformed into a question with a manually pre-defined template $Q : s_i \to q_i$. The overhead of creating these templates is minimal as it only has to be done once and is as simple as transforming the `name` slot in the `hotel` domain to a natural text question equivalent, e.g. *"What is the name of the hotel that the user wants?"*. Thus, with dialogue history until turn $t$ as $H_t = \{u_1, b_1, ..., u_{t-1}, b_{t-1}, u_t\}$, where $u_i$ is the user's utterance on the $i$th turn and $b_i$ is that of the bot's, the original single input output pair of

$$H_t \oplus T \to \{(s_{t,i} = v_{t,i}) \mid i \in I\} \quad (1)$$

becomes $N_T$ granular question answer pairs:

$$\{Q(s_{t,i}) \oplus H_t \to v_{t,i} \mid i \in I\} \quad (2)$$

where $\oplus$ denotes simple text concatenation. A difference from the original TransferQA approach is that since we will be finetuning the model, we skip the step of training with external question answering datasets and do not take any special measures to handle `none`, i.e., empty slots, because our models will learn to generate `none` as the answer for these slots. Further detail on the TransferQA format and additional examples of the fully constructed inputs are shared in Appendix A.1.

## 2.2 Fine-tuning with in-context examples

Adapting to new services can be made even more seamless by providing in-context examples (Wang et al., 2022; Min et al., 2022; Ouyang et al., 2022). Even when faced with a question it has never seen before, the examples provide guidance on how it should be answered. This kind of task reformulation enables the development of models that achieve state-of-the-art zero-shot performance and generalizability even with small models (60M parameters) by explicitly fine-tuning with instructions and in-context examples. Since most recent work that focus on generalizabilty and zero-shot models leverages generation models because of their open

vocabulary, we also place our focus on generation models.

Motivated by the results from Tk-instruct (Wang et al., 2022) and MetaICL (Min et al., 2022) that showed even relatively small models can generalize well if explicitly trained to follow instructions with examples, we explore whether we can prevent a model from overfitting to domain-specific questions and instead continually develop example-based question answering capabilities to enhance continual learning performance. Therefore, we extend Equation 2 to include in-context examples that are retrieved from the training set, as shown in (2) in Figure 2. To retrieve relevant examples, we use $H_t$ to form a query that retrieves the top $k$ samples $\{H_{t'}^{'j} | j \leq k\}$ to use as in-context examples.[3] By inserting the retrieved examples and their relevant slot values for each slot question $q_i$, the final format becomes:

$$\{Q(s_{t,i}) \oplus \{H_{t'}^{'j} \oplus v_{t',i}^{'j} | j \leq k\} \oplus H_t \rightarrow v_{t,i} \mid i \in I\} \tag{3}$$

Throughout this work, we use $k = 1$ unless otherwise specified.

### 2.3 Retrieving relevant in-context examples

The goal of the retrieval system is to find an example turn $H_{t'}'$ that requires similar reasoning for answering the target sample $H_t$, such that fine-tuning with it as an in-context example will help enable the model to apply the same reasoning for answering the question for the target sample. Hu et al. (2022) found that instead of matching for dialogue state overlap, matching for similar dialogue state change $\Delta DS$, i.e. state change similarity (SCS), yields more relevant examples. State changes are simply a subset of $DS$ that is different from the previous turn: $\Delta DS = \{(s_{t,i}, v_{t,i}) \mid i \in I, v_{t,i} \neq v_{t-1,i}\}$.

We found that computing similarity with this definition of state change results in many ties that leads to less relevant examples being lumped into the same rank as more relevant ones, so we make minor modifications by including the $\Delta DS$ operations, e.g. INSERT, DELETE, and UPDATE, as part of the slot key: $\Delta DS_{ours} = \{(s_1 \oplus o_1, v_1), ...(s_m \oplus o_m, v_m)\}$, where $o$ is the slot operation. To resolve ties that still remain with this modification, we use the BM25 (Robertson et al., 2009) score between the target and example's last bot and user utter-

ances $(b_t - 1, u_t)$.[4] With our changes, we were able to observe a much better top $k = 1$ match, which we verified manually with 100 random samples. We denote examples retrieved with this new SCS+BM25 score as the *Oracle* because getting $\Delta DS$ requires knowing the DS that we would like to predict ahead of time, and therefore cannot be used at test time. However, the Oracle score is useful for training a retriever that can retrieve examples with similar $\Delta DS$ and for estimating the upper bound for DST-EGQA.

Using the Oracle score, for each sample in the training set, we calculate its similarity with other training samples and select the top 200 samples. From the selected samples, we pair the top ten and bottom ten as hard positive and hard negative samples, respectively, to train a SentenceBERT-based (Reimers and Gurevych, 2019) retriever using contrastive loss. We call the resulting retriever IC-DST-retriever v2 (**IDR2**). This is the same configuration for creating the dataset that was used to train the original retriever used for IC-DST, but instead of using $x\%$ of the entire training data, we use the entire training set of the first domain $T_1$ to train separate retrievers for each of the five domain orderings. We impose this constraint such that we conduct our experiments under the practical assumption that we are only provided data for $T_1$ at the beginning and we do not want to extend the continual learning problem for training the retriever. More details of IDR2's training procedure can be found in Section A.3.

### 2.4 Dialogue-level sampling for memory

The approaches that we outlined thus far are not orthogonal to existing continual learning methods. Therefore, they can be combined to further boost performance. One of the simplest methods is memory replay, which samples training data from previous tasks and adds them to the current training set so that the models forget less. For memory replay to be effective, it is important to select representative and nonredundant training samples.

In DST, a training sample is a single turn in a dialogue, since dialogue state is predicted for every turn. To reduce redundant instances, we propose a simple change to selecting training samples. Instead of combining turns from all dialogues and then randomly sampling turns, we propose sam-

---

[3]Note that $t \neq t'$ because the retrieved example may not occur at the same $t$th turn.

[4]Refer to Appendix A.2 for the details of the original definition of state change similarity and the reasoning behind our modification details.

pling at the dialogue-level first and then including all turns from the sampled dialogues to form the memory. The motivation is that there are rarely the same type of dialogue state updates within a dialogue, but there is a high chance that frequent dialogue state updates across dialogues may be sampled multiple times when using turn-level sampling.

The simple difference between the sampling strategies are clearer when observing their code snippets in Python 3:

**Turn-level sampling.**

```python
# flatten() turns a nested list into a
    single-level list.
chosen_turn_samples = random.sample(
    flatten(dialogue), memory size)
```

**Dialogue-level sampling.**

```python
samples = random.sample(dialogue,
    memory size//10);
chosen_turn_samples = random.sample(
    flatten(samples), memory size)
```

## 3  Experimental Setup

### 3.1  Data

We use the continual learning setup proposed by Zhu et al. (2022), which uses 15 single domains from the Schema Guided Dialogue dataset (Rastogi et al., 2020), and aggregate our results over the same five domain orders to make the most reliable comparisons with their results. Comparing results with the same order is crucial as we find that results can have significant variance depending on the chosen domains and their order. For multi-task training, there is only a single permutation, and therefore we aggregate results over runs with three different seed values. Our formulation described in Section 2.2 shows that we are operating under the assumption that the domain of interest will be known ahead of time.

### 3.2  Evaluation

DST performance is mainly measured by joint goal accuracy (JGA), which indicates the percentage of turns for which all slot values are correctly predicted. For CL, given JGA for domain $i$ after training up to the $t^{\text{th}}$ domain $a_{t,i}$ and the total number of domains $T$, we compare our approaches with three metrics from Zhu et al. (2022):

*(i)* Average JGA $= \frac{1}{T} \sum_{i=1}^{T} a_{T,i}$, the average of JGA on each domain after training on all domains in the continual learning setup, *(ii)* Forward Transfer (FWT) $= \frac{1}{T-1} \sum_{i=2}^{T} a_{i-1,i}$, how much training on the current domain boosts JGA on future unseen domains, and *(iii)* Backward Transfer (BWT) $= \frac{1}{T-1} \sum_{i=1}^{T-1} a_{T,i} - a_{i,i}$, how much the training on the current domain reduces JGA on data from previously seen domains. We place the most importance on Final JGA, while FWT and BWT provide additional signal on how different approaches provide more transferability, and hence task consistency, between domains.

### 3.3  Baselines

We replicate the baseline results from Zhu et al. (2022) using their implementation, which include approaches from Madotto et al. (2021):

- SimpleTOD (Hosseini-Asl et al., 2020): perform DST as a structured text generation task, predicting the full state as a single sequence. As was done in Zhu et al. (2022), we modify the SimpleTOD format to append the domain name at the end of the dialogue history as described in Equation 1.

- Memory: randomly select $M$ turns from the training data for each previous domain and include it in the current domain's training data.

- EWC: use the same samples selected for memory replay to regularize with the Fisher information matrix (Kirkpatrick et al., 2017)

- AdapterCL (Madotto et al., 2021): freeze the base model and train parameter efficient adapters for each domain with number of weights that are equivalent to 2% of that of the pretrained model.

- Continual Prompt Tuning (Zhu et al., 2022): freeze the base model and continually train soft prompts after reformulating DST as a masked-span recovery task (Raffel et al., 2020). We include their best results, which take advantage of a memory buffer for replay and for memory-guided backward transfer, a form of regularization that prevents updates if doing so would increase the current model's loss on the memory samples by computing gradients on them.

For DST-EGQA, we compare various configurations to better understand the strengths and weak-

nesses of our approach. We vary the retriever used during training and combine with other memory replay strategies. We also show CPT Multi-task and DST-EGQA Multi-task to show the multi-tasking upper bound performance for average JGA.

### 3.4 Retrieval baselines

We also experiment with various retrieval techniques as baselines to IDR2:

- Random sampling
- BM25 (Robertson et al., 2009)
- OpenAI's `text-embedding-ada-002` model[5]
- SentenceBERT (Reimers and Gurevych, 2019): the `all-mpnet-base-v2`[6]
- The original IC-DST retriever (oIDR): the retriever from Hu et al. (2022) that was trained with the original SCS formulation and pairs created from the MultiWOZ 2.1 dataset (Eric et al., 2020).

Other than random sampling and BM25, retrieval ranking is based on the similarity between sentence embeddings, which is the dot product between the query and the key. With the exception of oIDR, which was trained to identify similarity with the last turn's dialogue state and last utterance pairs between the bot and user: $\{(s_{t-1,i} = v_{t-1,i}) \mid i \in I\} \oplus u_{t-1} \oplus u_t$, the query and key of the database uses only the last utterance pairs: $u_{t-1} \oplus u_t$. We found this approach to be better as it diminishes the undesirably high similarity assigned to examples from the same dialogue that have the same previous dialogue state.

### 3.5 Technical details

We conduct our experiments with the T5-small model (Raffel et al., 2020). We train with a single GPU using the AdamW optimizer, a learning rate of 1e-4, and a batch size of 16. We train on each domain for ten epochs without early stopping. We select the checkpoint with the best validation set performance when moving on to the next domain. Our experiments are run on V100, A40, and A100 GPUs, based on availability.[7]

---

[5]https://platform.openai.com/docs/guides/embeddings
[6]https://www.sbert.net/docs/pretrained_models.html
[7]Our preliminary experiments with different GPU types with otherwise identical configurations showed that the choice

## 4 Experiments and Analysis

### 4.1 Main results

**TransferQA's format is more CL-friendly.** The results for only transforming the DST from prior work (Equation 1) to that of granular question answering using the TransferQA (Equation 2) format is shown in the row for DST-EGQA − In-context examples in Table 1. Without the help of any in-context examples, the transformation alone yields a dramatic improvement in CL performance, increasing average JGA from 14.4 to 43.2, and also improving on both FWT and BWT. These results supports our hypothesis that a question answering task that is understandable through natural language is more conducive to better continual learning than learning to generate service-specific structured output.

**Example-guided question answering further enhances CL performance.** The subsequent rows for DST-EGQA shows that fine-tuning with in-context examples can further enhance all CL metrics by a large margin. Most notable is the boosts that are seen in the FWT, for which memory replay has almost a negligible effect. Augmenting DST-EGQA with memory replay leads to even larger boosts, even exceeding the CPT Multi-task model, with most gains coming from BWT, which is expected with memory replay methods. Using the Oracle retriever at test time leads to statistically insignificant improvements, indicating that IDR2 can retrieve examples that are on par with the Oracle examples. Lastly, we can see that the relative gains in Average JGA and BWT from memory replay becomes less pronounced with models trained with in-context examples, indicating that memory replay and example-guided question answering have overlapping gains.

**Double-dipping the training set as a retrieval database does not lead to overfitting.** It is important to note that, because our retrieval methods are commutative, a target sample that is paired with an example will serve as an example when the example becomes the target sample. Therefore, the answers for all training samples are seen as part of the context during training with our setup described in Section 2.3. This raises overfitting concerns that the model could easily memorize the

---

of GPU in final performance introduces minimal variability to the final result.

| Method | Retriever | Avg. JGA | FWT | BWT | +Memory | +Params | +Reg. |
|---|---|---|---|---|---|---|---|
| SimpleTOD (2020) | | $14.4_{2.7}$ | $7.1_{1.0}$ | $-42.5_{2.4}$ | - | - | - |
| EWC (2017) | | $13.9_{1.1}$ | $8.4_{0.9}$ | $-50.8_{4.3}$ | ✓ | ✓ | ✓ |
| Memory (2021) | - | $58.6_{3.5}$ | $10.9_{0.5}$ | $-3.2_{2.3}$ | ✓ | - | - |
| Adapter (2021) | | $49.8_{1.7}$ | - | - | - | ✓ | - |
| CPT + Memory (2022) | | $61.2_{2.5}$ | $13.7_{0.8}$ | $\mathbf{0.5_{0.4}}^{\dagger}$ | ✓ | ✓ | ✓ |
| DST-EGQA | IDR2 | $54.1_{3.3}$ | $\mathbf{22.8_{1.8}}$ | $-22.3_{4.5}$ | - | | |
|   + Dialogue Memory | | $\mathbf{68.9_{0.3}}^{\dagger}$ | $21.2_{1.5}$ | $-6.1_{1.7}$ | ✓ | - | - |
|   − In-context examples | - | $43.2_{3.4}$ | $14.1_{1.9}$ | $-31.0_{4.2}$ | - | | |
| DST-EGQA | Oracle | $55.5_{3.5}$ | $23.6_{2.1}$ | $-19.1_{4.2}$ | - | | |
|   + Dialogue Memory | | $69.3_{1.0}$ | $22.5_{1.8}$ | $-5.9_{1.9}$ | ✓ | - | - |
| CPT Multi-task (2022) | - | $64.0_{1.9}$ | - | - | - | ✓ | ✓ |
| DST-EGQA Multi-task | - | $74.2_{1.8}$ | - | - | - | - | - |

Table 1: CL metric results with a checklist on the reliance of other continual learning techniques. We compare models sequentially trained on 15 tasks from the SGD dataset and aggregate results across five different domain permutations. DST-EGQA achieves the best results without any additional parameters or regularization methods. The last two rows provide the multi-tasking results, which serve as an upper bound. In this table, results with retrievers are with a single in-context example and the indicated retriever is used for training and test time, while the Oracle retriever is used for the validation set. Memory here refers to samples that are added for the training data of subsequent services for memory replay. All rows that use memory are with memory budget of $M = 50$. $^{\dagger}$ indicates statistically significant at $p < 0.05$ with the next best comparable value.

| Train | Dev | Test | Avg. JGA | FWT | BWT |
|---|---|---|---|---|---|
| - | - | - | $43.2_{3.4}$ | $14.1_{1.9}$ | $-31.0_{4.2}$ |
| IDR2 | IDR2 | | $45.1_{3.0}$ | $21.4_{1.4}$ | $-31.9_{3.4}$ |
| IDR2 | Oracle | IDR2 | $\mathbf{54.1_{3.3}}^{\dagger}$ | $\mathbf{22.8_{1.8}}$ | $\mathbf{-22.3_{4.5}}$ |
| Oracle | Oracle | | $48.5_{3.2}$ | $19.6_{1.6}$ | $-27.1_{1.4}$ |
| Oracle | Oracle | Oracle | $53.7_{4.4}$ | $24.1_{2.6}$ | $-21.3_{4.1}$ |
| IDR2 | | | $55.5_{3.5}$ | $23.6_{2.1}$ | $-19.1_{4.2}$ |

Table 2: Train-validation-test retrieval method comparison. Keeping the Training and Test-time retrieval methods the same while keeping the development set as the Oracle leads to the best results, except for the last row, which requires knowing the correct answer ahead of time. $^{\dagger}$ indicates statistically significant at $p < 0.05$ with the next best value.

answers for all samples and thus not learn generalizable question-answering. Interestingly, this does not seem to be the case, as training in this setup leads to improved or on-par final test set performance compared to training without any examples. This implies that our approach does not impose additional data constraints of having to split the training set into dedicated training samples and retrieval samples for it to be effective.

However, not shown in Table 1 is that we find that DST-EGQA is sensitive to the training dynamics (Section 4.2) and the quality of the retrieved examples (Section 4.3).

## 4.2 Training dynamics

In practical settings we don't have an oracle retriever, and our database may not contain the a perfect example for each case seen at test time. Thus, we may in fact retrieve irrelevant examples. It is important for the model to be able to handle these situations. Specifically, it should be able to leverage relevant examples, yet ignore irrelevant ones. To become more robust to these realistic circumstances, it may be useful to intentionally mix in irrelevant examples during training for DST-EGQA. We vary the combination of IDR2 and Oracle used for training, validation, and test time. Results in Table 2 support our hypothesis, showing that aligning the retrieval method from training time with the method used at test time leads to the best performance. Interestingly, best performance is achieved by using the Oracle retriever at validation time, shown by the large gap between IDR2 → IDR2 → IDR2 and IDR2 → Oracle → IDR2 (second and third row). This is somewhat surprising given that one may expect selecting a checkpoint that performs the best in the same setting as test time would lead to better test time performance.

## 4.3 Retrieval method sensitivity

The findings from Section 4.2 raises a question on whether training with other retrievers that may provide a different mixture of good and bad exam-

| Train, Test | Dev | Avg. JGA | FWT | BWT |
|---|---|---|---|---|
| - | - | $43.2_{3.4}$ | $14.1_{1.9}$ | $-31.0_{4.2}$ |
| Random | | $45.5_{4.5}$ | $14.2_{2.2}$ | $-31.4_{5.1}$ |
| BM25 | | $46.7_{3.3}$ | $21.6_{1.6}$ | $\mathbf{-20.1_{5.0}}$ |
| SentBERT | Oracle | $46.2_{6.0}$ | $17.3_{2.0}$ | $-29.7_{6.9}$ |
| GPT | | $47.8_{7.9}$ | $17.5_{2.4}$ | $-27.0_{8.7}$ |
| oIDR | | $49.2_{4.7}$ | $19.9_{2.1}$ | $-26.2_{5.2}$ |
| IDR2 (ours) | | $\mathbf{54.1_{3.3}}^{\dagger}$ | $\mathbf{22.8_{1.8}}$ | $-22.3_{4.5}$ |

Table 3: Retrieval methods comparison. Although mixing in irrelevant examples can boost performance at training time, our results show that lacking a reliable retrieval method at test time is detrimental to performance. Our IDR2 model captures this balance the most effectively.

| Size | Method | Avg. JGA | FWT | BWT |
|---|---|---|---|---|
| - | - | $43.2_{3.4}$ | $14.1_{1.9}$ | $-31.0_{4.2}$ |
| 10 | Turn | $50.1_{3.8}$ | $15.0_{1.4}$ | $-23.7_{4.4}$ |
| | Dialogue | $\mathbf{59.1_{1.5}}^{\dagger}$ | $\mathbf{15.2_{2.7}}$ | $\mathbf{-14.7_{2.3}}^{\dagger}$ |
| 50 | Turn | $59.8_{1.6}$ | $\mathbf{15.6_{1.7}}$ | $-12.8_{2.0}$ |
| | Dialogue | $\mathbf{64.2_{0.8}}^{\dagger}$ | $15.0_{2.1}$ | $\mathbf{-7.4_{2.2}}^{\dagger}$ |
| 100 | Turn | $63.9_{1.2}$ | $\mathbf{15.6_{1.7}}$ | $-8.7_{1.3}$ |
| | Dialogue | $\mathbf{66.8_{1.5}}^{\dagger}$ | $15.4_{2.1}$ | $\mathbf{-3.3_{2.5}}^{\dagger}$ |

Table 4: Memory size analysis for DST-EGQA. Sampling at the dialogue-level is much more effective than sampling at the turn-level, especially for a constrained memory budget.

ples can lead to a further boost performance with DST-EGQA. We apply all the retrievers defined in Section 2.3 and use the same training dynamics that led to best results previously to examine each retriever's effectiveness. As shown in Table 3, our IDR2 model seems to capture this balance the most effectively, as it is significantly better than all other retrieval methods.

## 4.4 Memory sampling strategy and size

We study the effect of the memory sampling strategy and the size of the memory budget. We do not use in-context examples for all configurations to study their effects in isolation. As hinted by the results in Table 1, dialogue-level sampling seems to be a superior sampling strategy to turn-level sampling. We take a deeper dive into the relationship between the two sampling techniques and how both approaches scale with memory budgets by varying the memory budget sizes to 10, 50, and 100. Here, size refers to the number of training samples. To make sure the comparison between turn-level and dialogue-level samples is fair, we sample dialogues until the total number of turns in sampled dialogues exceed the target size, and then sample the targeted

| Train, Test | # Ex. | Avg. JGA | FWT | BWT |
|---|---|---|---|---|
| - | - | $43.2_{3.4}$ | $14.1_{1.9}$ | $-31.0_{4.2}$ |
| Random | 1 | $43.2_{6.8}$ | $14.5_{1.7}$ | $-33.3_{5.7}$ |
| | 2 | $\mathbf{45.2_{4.5}}$ | $15.5_{1.8}$ | $-31.6_{6.2}$ |
| | 3 | $43.9_{6.2}$ | $\mathbf{16.9_{1.6}}$ | $\mathbf{-31.4_{6.7}}$ |
| BM25 | 1 | $45.9_{4.5}$ | $20.3_{1.9}$ | $-21.4_{6.2}$ |
| | 2 | $46.2_{6.1}$ | $\mathbf{23.3_{1.6}}$ | $\mathbf{-17.1_{7.5}}$ |
| | 3 | $\mathbf{47.0_{5.3}}$ | $20.8_{2.0}$ | $-21.8_{5.5}$ |
| IDR2 | 1 | $\mathbf{54.1_{3.3}}$ | $\mathbf{22.8_{1.8}}$ | $\mathbf{-22.3_{4.5}}$ |
| | 2 | $50.2_{3.7}$ | $22.0_{1.8}$ | $-29.3_{5.2}$ |
| | 3 | $48.0_{4.4}$ | $21.8_{1.9}$ | $\mathbf{-22.3_{4.1}}$ |
| Oracle | 1 | $53.7_{4.4}$ | $24.1_{2.6}$ | $-21.3_{4.1}$ |
| | 2 | $\mathbf{54.3_{3.0}}$ | $28.7_{2.6}$ | $-18.5_{3.6}$ |
| | 3 | $53.9_{3.8}$ | $\mathbf{30.5_{1.5}}$ | $-14.1_{2.6}$ |

Table 5: Number of in-context examples analysis. Small models are unable to leverage more than one in-context example when explicitly finetuned to perform in-context learning.

number of samples from the exceeded set.

Table 4 shows that dialogue-level sampling achieves a significantly better performance for all equivalent memory budget sizes for turn-level sampling and is even on par with the next budget size used for turn-level sampling. This is likely due to dialogue-sampling leading to a more comprehensive set of samples that cover a wider diversity of dialogue state updates in these smaller sizes of the memory budget as described in subsection 2.4. As the memory budget becomes larger, however, the gap between turn-level sampling and dialogue-level sampling diminishes, since both methods converge to multi-task training when the memory budget is unlimited.

## 4.5 Number of in-context examples

We also study the effect of having more than one in-context example and share the results in Table 5. Including only one example to learn from in-context creates a single point of failure, which is especially risky for suboptimal retrieval methods. Having additional examples to learn from can help mitigate this risk. Therefore, we repeat our experiments using multiple in-context examples. However, at least with small model sizes, the DST models are not able to effectively leverage additional examples. This is not surprising for the Oracle retriever, where in most cases the top example is the best example that can be leveraged from the training set.

## 5 Related Work

**Continual learning**   Continual learning prolongs the lifetime of a model by training it further with new incoming data without incurring the cost of

*catastrophic forgetting* (McCloskey and Cohen, 1989; French, 1999). There are three main branches of continual learning: architecture-based methods, replay-based methods, and regularization-based methods. Architecture-based methods propose dynamically adding model weights when learning new data (Fernando et al., 2017; Shen et al., 2019). Replay-based methods mitigate catastrophic forgetting by keeping a small sample of the previous data as part of a memory budget to train with the new data (Rebuffi et al., 2017; Hou et al., 2019). These methods mainly experiment with sampling strategies and memory budget efficiency. Lastly, regularization-based methods place constraints on how the model becomes updated during training with the new data such that its performance on previous data is maintained (Kirkpatrick et al., 2017; Li and Hoiem, 2018).

**Dialogue state tracking**  Continual learning for DST has been explored by a series of recent work that applied a combination of methods mentioned above. Liu et al. (2021) expanded on SOM-DST (Kim et al., 2020) with prototypical sample selection for the memory buffer and multi-level knowledge distillation as a regularization mechanism. Madotto et al. (2021) applied various continual learning methods to end-to-end task-oriented dialogue models and found that adapters are most effective for the intent classification and DST while memory is most effective for response generation. More recently, Zhu et al. (2022) proposed *Continual Prompt Tuning* (CPT), which is most related to our work. CPT improves continual learning performance by finetuning soft prompts for each domain and reformulating DST to align with T5's masked-span recovery pretraining objective (Raffel et al., 2020). Compared to CPT, we suggest a more granular reformulation to facilitate the learning from examples and do not rely on any regularization nor additional weights.

**Task reformulation and in-context learning** Enhancing a model's generalizability to various tasks by reformulating input and/or outputs to become more uniform has become an increasingly popular method for massive multi-task learning (Aghajanyan et al., 2021), even for tasks that are considered distant from one another. T5 (Raffel et al., 2020) accelerated this movement by providing dataset or task-specific labels or minimal instructions to the inputs and then doing multi-task

training. Building on T5, Sanh et al. (2022) and Wei et al. (2021) used more elaborate and diverse set of instruction templates and showed that this can significantly boost zero-shot performance. Cho et al. (2022) applied a similar idea to a more selective set of pre-finetuning tasks before training on the target DST dataset to improve DST robustness. Tk-instruct (Wang et al., 2022) takes a step further by scaling up the amount of tasks included in T0 and also provides positive and negative examples in the context in addition to the instructions. Similarly, Min et al. (2022) introduced MetaICL, which explicitly trains a model with the few-shot in-context learning format used for large language models (Brown et al., 2020), and showed that it showed better in-context learning performance than larger models. Task reformulation has also been recently explored to help the model better understand the task at hand and reduce domain-specific memorization and thus boost zero-shot DST performance (Li et al., 2021; Lin et al., 2021; Gupta et al., 2022; Zhao et al., 2022).

## 6 Conclusion

In this paper, we propose *Dialogue State Tracking as Example-Guided Question Answering* as a method for enhancing continual learning performance that factors dialogue state tracking into granular question answering tasks and fine-tunes the model to leverage relevant in-context examples to answer these questions. Our method is an effective alternative to existing continual learning approaches that does not rely on complex regularization, parameter expansion, or memory sampling techniques. Analysis of our approach reveals that even models as small as 60M parameters can be trained to perform in-context learning for continual learning and that complementing such a model with a randomly sampled memory achieves state-of-the-art results compared to strong baselines.

## Limitations

Using the TransferQA idea and retrieved examples for in-context fine-tuning adds a lot of configurations, which we have not been exhaustively explored, in lieu of prioritization of, we judged, more important experiments. For example, we did not explore sensitivity to the specific wording of questions, as was done with T0 (Sanh et al., 2022). We leave as future work the testing of the hypothesis that having more diverse questions per slot can lead

to even more generalizability between domains and bring even further improvements to DST-EGQA.

Another limitation of DST-EGQA is that the retrieval database stores all previously seen samples from training and thus can be considered a memory with infinite size in our current formulation. Although the samples in the retrieval database are not used as training samples and provided as in-context examples during inference after being trained with subsequent services, the memory requirement for maintaining the database may be quite high. However, we believe that this memory requirement is still less restrictive than having the compute for fully retraining the model with all data whenever the model needs to learn a new service, especially when the training data set is large.

Lastly, an important practical consideration is the varying technical overhead in implementation and portability of different approaches. Compared to other approaches, training and inference is relatively simple, as we use an autoregressive text generation objective without special modifications. However, while our approach does not require any additional parameters, it does require a database and a retrieval model that is comparable in size to the DST model. Therefore, depending on the technical constraints, managing these two components may be less desirable.

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

# Appendix

## A  Additional details

### A.1  TransferQA format

The TransferQA format (Lin et al., 2021) has two different formats depending on whether the question is multiple choice or extractive. Categorical questions with fixed answer choices are given as multiple choice questions where the options are provided as part of the context, prepended by [opt]. Extractive questions do not have a special format. We remove the prefix Extractive Question: and Multi-choice QA as the presence of [opt] already allows the model to distinguish between the two. In addition, when there are in-context examples, we add [target] and [example] so that the model can distinguish between the examples and the target. The original and our modifications are shown in Figure 3.

### A.2  State change similarity

Given two sets of state changes represented as $\Delta DS_a = \{(s_1, v_1), ...(s_m\, v_m)\}$ and $\Delta DS_b = \{(s_1, v_1), ...(s_n\, v_n)\}$, where $s$ is the slot key and $v$ is the slot value to update the slot to, Hu et al. (2022) defines state change similarity (SCS) as the average of the similarity between slot keys $s$, $F_{slot}$ and the similarity between key value pairs, $F_{slot-value}$:

$$SCS(\Delta DS_a, \Delta DS_b) = \frac{1}{2}(F_{slot} + F_{slot-value})$$

Similarity $F$ between the two sets is measured by computing the average of two $F_1$ scores using each set as the target. We use the standard calculation: $F_1 = \frac{2PR}{P+R}$, where $P$ is precision and $R$ is recall.

The resulting ties with this formula were not as critical for IC-DST (Hu et al., 2022), because top $k$ examples, where $k$ is a sizeable value that includes most ties, were all provided as in-context learning examples.

### A.3  IC-DST retriever v2

We use the same hyperparameter settings as oIDR (Hu et al., 2022), which uses a learning rate of $2e^{-5}$, 1000 warm-up steps, and the contrastive loss objective. We experimented with a binary classification and triplet evaluator at test time for discriminating between similar and dissimilar samples and selecting the best checkpoint to use. The binary classification evaluator determines accuracy based on cosine similarity between the paired examples and an automatically calculated similarity threshold that results in the best accuracy. The triplet evaluator, on the other hand, compares whether $distance(q, p) > distance(q, n)$, where $p$ is the similar pair, $n$ is the negative pair, and $q$ is the query that serves as the anchor between the similar and dissimilar samples. We find that the two evaluators yield statistically insignificant differences in the final performance for our approach and therefore we use the triplet evaluator for all the results included in this work. We exclude the previous turn's dialogue state as the input query when fine-tuning SentBERT (Reimers and Gurevych, 2019).

**Extractive QA**

**Multi-choice QA**

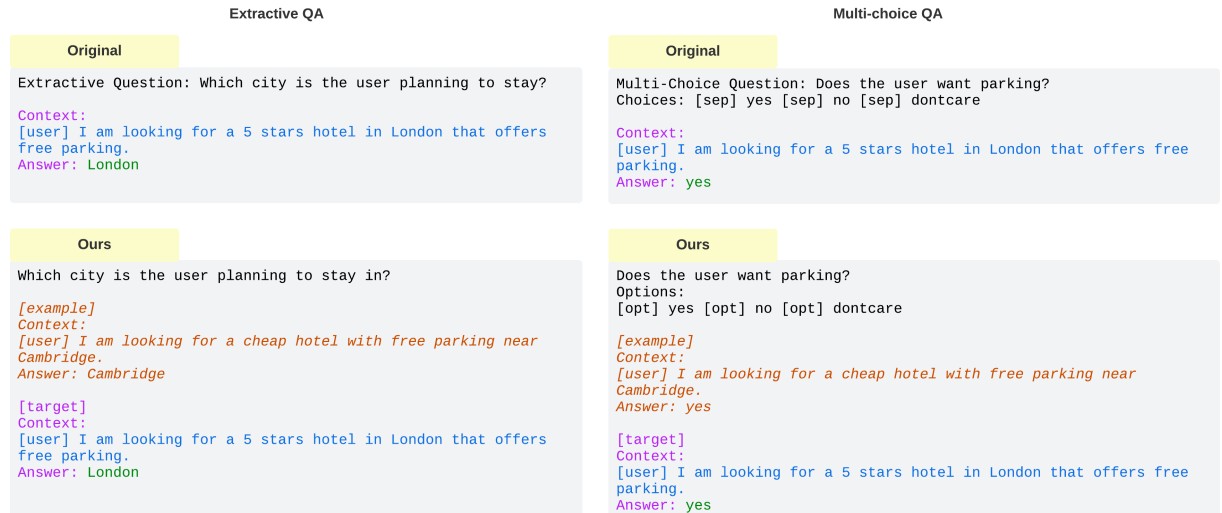

Figure 3: An illustration of the original and modified version of the TransferQA format. The desired output is in green, which is the text that comes after the last `Answer:`.