# OpenReview forum: "Continual Dialogue State Tracking via Example-Guided Question Answering"
_EMNLP/2023/Conference — EMNLP 2023 Main_

### Official Review · Reviewer_GgpV · 2023-07-25

**Soundness:** 3

**Excitement:**

3: Ambivalent: It has merits (e.g., it reports state-of-the-art results, the idea is nice), but there are key weaknesses (e.g., it describes incremental work), and it can significantly benefit from another round of revision. However, I won't object to accepting it if my co-reviewers champion it.

**Paper Topic And Main Contributions:**

This paper proposes alleviating the catastrophic forgetting problem in continual learning by formulating the Dialog State Tracking (DST) task as an Example-Guided Question Answering (EGQA) task, which reduces the data distribution gap between domains and takes advantage of in-context learning. The authors transform DST to QA like TransferQA (Lin et al., 2021), and retrieve similar samples from the training set as in-context examples like IC-DST (Hu et al., 2022). Experiments on continual learning DST show the effectiveness of EGQA reformulation and analyze method robustness on different retrievers, memory types and sizes, and the number of in-context learning examples.

**Questions For The Authors:**

See "Reasons To Reject".

**Reasons To Accept:**

1. The idea of reformulating DST to EGQA is effective for continual learning, which not only alleviates the catastrophic forgetting problem but also improves model generalization on unseen domains. Utilizing retrieval via in-context learning may also work well on tasks other than DST in the continual learning setting.
2. The authors conduct detailed experiments to analyze the effects of different factors, such as the types of retrievers and the number of in-context learning samples.


**Reasons To Reject:**

1. The main contribution of this paper is reformulating DST as EGQA. However, the process of retrieving in-context examples is not clear. I guess the database consists of all samples of the current domain (i.e., the domain of the target dialog), and the IC-DST-ret retriever compares the last utterance pairs ($u_{t-1}$, $u_t$) of $H_t$ and $H_t^\prime$ (To the authors: please correct me if I'm wrong). In this way, EGQA needs to store all samples from all domains in the database, which should be regarded as using **infinite memory** (instead of "None" in Table 1) since all samples from previous domains are available. That would be a disadvantage of the proposed method. It can be improved by limiting the number of samples of each slot/domain in the database, like the memory replay method.
2. The authors "use the entire training set of the first domain" to train the retriever and then use the retriever for all subsequent domains. The authors claim that "we do not want to extend the continual learning problem for training the retriever." However, could that retriever work well on other domains? More evaluation and discussion are appreciated.
3. "Dialogue-level sampling" is unexplained. As the authors claim it is one of the main contributions, it should be discussed explicitly.

**Reproducibility:**

3: Could reproduce the results with some difficulty. The settings of parameters are underspecified or subjectively determined; the training/evaluation data are not widely available.

**Reviewer Confidence:**

4: Quite sure. I tried to check the important points carefully. It's unlikely, though conceivable, that I missed something that should affect my ratings.

**Typos Grammar Style And Presentation Improvements:**

Line 183: "we will be finetuning the model" => "we will finetune the model";

Line 380: The number "43.2" should refer to Table 2.

---

> ### Author Rebuttal · Authors · 2023-08-28
>
> Thank you for your thorough review of our work and for appreciating the core contributions in restructuring the DST task to example-guided question answering to improve continual learning performance. You have raised important questions that will greatly improve the clarity of our paper:
>
> - Infinite memory:
>     - You are correct in that the retrieval database can be considered an infinite memory as the training samples are kept for retrieving examples at inference time. Therefore `+Memory = None` is misleading for the row with results of DST-EGQA without memory. We wanted to indicate that there are no additional **training** samples from previous domains. We will clarify this by creating separate columns for `+Memory` and `+Compute`. Considering that memory is cheap while compute is not, keeping samples for retrieval is a very minor disadvantage while keeping a large memory for training samples from previous domains to retrain on pushes the setting closer to full retraining at each new domain, which is antithetical to continual learning. However, we may make a stronger case for our approach if we see performance gains remain intact even after limiting the size of the retrieval database per domain. We will include this analysis in our revision.
>     - The IC-DST-ret retriever uses the previous turn’s dialogue state and also the previous bot utterance and user utterance ($\{(s_{t-1,i}=v_{t-1,i})\mid i \in I\} \oplus u_{t-1} \oplus u_t$), instead of just the last user utterance pairs for matching similarity. This is explained towards the end of Section 2.3 in lines 278-279, but will make this information more clear as it is an important detail.
>
> - Generalizability of retriever across domains:
>     - Our setup to train the retriever only on the first domain specifically serves to answer your question on whether such a retriever can still be useful for the subsequent domains that it has not yet seen. Our results, which incorporate five different domain order variations from [Zhu et al. 2022](https://aclanthology.org/2022.acl-long.80/), show that it can be. If we extend continual learning setup to the retriever as well, as in continually updating it for each new domain it encounters, we believe the performance of DST-EGQA will be enhanced further, but it introduces additional overhead in managing multiple models for continual learning. We will include a table on the specific domain ordering in the appendix for additional clarity.
>
> - Dialogue-level sampling is unexplained:
>     - Thank you so much for pointing out this oversight! Dialogue-level sampling for memory replay as opposed to turn-level sampling is that samples are chosen at the dialogue-level first and turns of the selected dialogues up to the memory size are put into memory. Pseudocode:
> `samples = random.sample(dialogue, memory size//10); chosen_turn_samples = random.sample(flatten(samples), memory size))`
>     - On the other hand, turn-level sampling means that the samples are flattened out to turns first and then randomly sampled from all turns (pseudocode: `samples = random.sample(flatten(dialogue), memory size`). Although this is a simple modification, we observe that this enables a wider diversity of dialogue state updates to be captured in smaller memory budgets without incorporating more complicated techniques to store prototypical samples.
>     - Your thorough review also let us realize that the performance gain of only using the TransferQA format was omitted from Table 1, making the numbers from the memory analysis table potentially confusing. We will clarify that the base number is for when only the TransferQA format is applied, i.e. without using any retrieved examples, and the following rows are only when various memory sampling methods are used. This lets us isolate the effect of memory from that of the retrieved examples.

---

### Official Review · Reviewer_88YD · 2023-07-26

**Soundness:** 3

**Excitement:**

4: Strong: This paper deepens the understanding of some phenomenon or lowers the barriers to an existing research direction.

**Paper Topic And Main Contributions:**

Paper topic: Continual learning for dialogue state tracking.
Main contributions:
(1) reformulate dialogue state tracking as a fine-grained example-guided question answering task.
(2) propose a novel dialogue-level sampling strategy for choosing memory samples.

**Questions For The Authors:**

Overall, this paper is well-motivated and novel, I tend to see its appearance in EMNLP, but there are still some points that can be improved:
1. I suggest that authors could try some larger models.
2. Table 2 is unreadable.
3. Some analysis on large models like ChatGPT can be included, even in Appendix. As this tells us how SOTA models performs on this task.

**Reasons To Accept:**

1. A novel perspective to view DST as a fine-grained example-guided question-answering task to improve continual learning performance.
2. Propose a simple dialogue-level sampling strategy for choosing memory samples in the experience-replay approach.

**Reasons To Reject:**

1. I suggest that authors could try some larger models like FlanT5/T5-large or so to test their continual learning approach for the dialogue state tracking problem.
2. The motivation of this paper should be better described, especially its sampling strategy.
3. The presentation, especially the tables and figures, can be improved.

**Reproducibility:**

3: Could reproduce the results with some difficulty. The settings of parameters are underspecified or subjectively determined; the training/evaluation data are not widely available.

**Reviewer Confidence:**

3: Pretty sure, but there's a chance I missed something. Although I have a good feel for this area in general, I did not carefully check the paper's details, e.g., the math, experimental design, or novelty.

---

> ### Author Rebuttal · Authors · 2023-08-28
>
> Thank you for your thorough review and for appreciating the core contributions of our work in restructuring the DST task to example-guided question answering to improve continual learning performance.
> - Larger models: We understand your interest to see larger models applied to this task and we will also include experiments with T5-large and Flan-T5 to shed light on the performance of larger models and also models that have been trained to follow instructions, which can be considered a more generalizable behavior of answering questions.
> - Models like ChatGPT are much larger and are used mostly for zero-shot scenarios due to the practical difficulty of fine-tuning them. Therefore, continual learning with ChatGPT-scale models will be challenging, but it would be interesting to compare its zero-shot performance using our setup and we will explore it. On a related note, [Heck et al. 2023](https://aclanthology.org/2023.acl-short.81/) suggests that it will likely achieve strong performance as it achieves similar results to IC-DST.

---

### Official Review · Reviewer_6XWq · 2023-08-04

**Soundness:** 3

**Excitement:**

4: Strong: This paper deepens the understanding of some phenomenon or lowers the barriers to an existing research direction.

**Paper Topic And Main Contributions:**

This work proposes to improve continual learning in DST by eliminating service-specific structured text and decomposing each DST sample to a list of fine-grained example-guided question answering tasks. Specifically, the classical way of DST is reformatted as a QA problem asking for the value of each slot. The model input includes the target dialogue, the slot question, example dialogue, and example slot value. The model then outputs the answer to the slot question. Examples were retrieved from the training data, or the database. Ideal samples are those that require similar reasoning for answering the target sample. This is done via measuring the state change similarity and BM25 re-ranker.

Authors compared DST-EQGA with a range of models using SGD dataset under a continual learning set-up. Authors also proposed a dialogue-level memory sampling strategy and compared with other sampling approaches. Overall, the proposed DST-EQGA framework achieved state-of-the-art continual learning performance with a T5-small model (60M) parameters.

**Reasons To Accept:**

The paper is well-written with thorough experimental results and analysis.

The proposed method, DST-EGQA, significantly improves the continual learning performance of DST on SGD dataset. DST-EGQA achieves state-of-the-art performance with a 60M model. This is quite remarkable especially in the era of large language models.

**Reasons To Reject:**

JGA of each domain should be included. It is difficult to tell is certain domains are considered easier than others, or if this method has difficulty in adapting to certain domain. Such analysis would be important for the continual learning scenario.

Only SGD is evaluated. More dataset (MultiWOZ, etc.) and even cross-dataset domains should be included to consolidate the findings.

**Reproducibility:**

4: Could mostly reproduce the results, but there may be some variation because of sample variance or minor variations in their interpretation of the protocol or method.

**Reviewer Confidence:**

3: Pretty sure, but there's a chance I missed something. Although I have a good feel for this area in general, I did not carefully check the paper's details, e.g., the math, experimental design, or novelty.

---

> ### Author Rebuttal · Authors · 2023-08-28
>
> Thank you very much for your thorough review and for appreciating the core contributions of our work in restructuring the DST task to example-guided question answering to improve strong continual learning performance even with a small T5 model. We would like to address some of your concerns:
> - JGA of each domain should be included: These are intermediate results that are necessary to calculate the final results already included in the paper and therefore a minor fix. We will make by including these results in the appendix with analysis on various difficulty levels of each domain as it may provide insight into even more improved continual learning performance. Fortunately,
> - Only SGD is evaluated: Our preliminary experiments were actually conducted with MultiWOZ following the setup from previous work ([Liu et al. 2021](https://aclanthology.org/2021.emnlp-main.176/)) and we observed similar results (50.05% average JGA even with only using TransferQA format vs 40.14% average JGA for KPN, the best performing model in the previous work) but we found the setup to involve a smaller set of domains that were not independent from one another (`hotel` vs `hotel + restaurant` vs `restaurant` were considered separate domains). This makes MultiWOZ a poor test bed for continual learning performance for DST and so we pivoted to the SGD setup by [Zhu et al](https://aclanthology.org/2022.acl-long.80). However, we will repeat some of these experiments with MultiWOZ and include them in the appendix of the revised version of our paper for those interested and mention the caveats of using this dataset.

---

### Meta-Review · Area_Chair_1ty6 · 2023-09-10

**Recommendation:** 4

**Metareview:**

This paper studies the "dialogue state continual learning" problem. The authors present a novel and straightforward example-guided QA solution, achieving SOTA on the SGD dataset. The solution is built on three foundational ideas:

1. **Reformulation of DST into QA (DST-EGQA)**: This approach considerably enhances continual learning performance by improving task consistency across domains.

2. **In-context Learning** by retrieving relevant in-context examples (from past domains) during both fine-tuning and testing. This method is akin to Memory-replay, given its exposure to past training examples. However, its performance (when combined with DST-EGQA as DST-EGQA + IC-DST-ret) doesn't measure up to Memory-replay (with a 50 turn-level example memory size). Despite this, the third pillar demonstrates that Memory techniques and In-context fine-tuning can collectively enhance the performance.

3. **Dialogue-level Memory-replay**: Sampling examples at the dialogue level proves superior to turn-level sampling. This is attributed to the richer dialogue state a dialogue offers compared to an individual turn.

While the original paper has moments of ambiguity in presenting its main contributions and experiments, the authors managed to clarify these areas during the rebuttal phase, providing reviewers and AC with a clearer understanding of their innovations. Nonetheless, there are instances of overstatements in the paper, and I recommend the authors reevaluate and modify their claims and experimental presentation.

Regarding the scores provided: Soundness scores are uniformly at (3, 3, 3), while excitement scores varied at (4, 4, 3). It's evident that all reviewers are enthused by the proposed solution due to its evident performance improvements in the experimental settings. Additionally, the simplicity of the proposed solutions is commendable.

However, reviewers expressed concerns about:

1. The single testing benchmark.
2. Limiting their experimentation to a single model size (60M).
3. A potentially skewed comparison with the Memory-replay method.

AC's feedback on these issues:

1. **Single Testing Benchmark**: Though multiple benchmarks typically provide a more comprehensive demonstration of a solution's generalizability, the SGD is arguably the most fitting benchmark for the proposed solution. This is because other dialogue benchmarks combine domains, which may not serve as the ideal "proof-of-concept" benchmark for this study. Authors might consider other benchmarks (like a sequence of mixed-domain tasks) to further fortify their experimental claims.

2. **Model Size**: In their rebuttal, the authors mention testing larger models and observing similar findings to the 60M size. But without presenting these specific results, this concern remains unresolved.

3. **Comparison with Memory-replay**: A reviewer pointed out a potential unfair in comparison with the Memory-replay method. However, after examining lines 333-335 of the paper, it appears the authors incorporated all past examples when using the Memory-replay method. Authors might want to clarify this for enhanced transparency.

In conclusion, this study comes across as a "proof-of-concept" endeavor, mostly justifying its contributions. I consider the work to be of **moderate sound and exciting**.

---

### Decision · Program_Chairs · 2023-10-07

**Decision:**

Accept-Main

**Comment:**

This paper studies the "dialogue state continual learning" problem. The authors present a novel and straightforward example-guided QA solution, achieving SOTA on the SGD dataset. The solution is built on three foundational ideas:

1. **Reformulation of DST into QA (DST-EGQA)**: This approach considerably enhances continual learning performance by improving task consistency across domains.

2. **In-context Learning** by retrieving relevant in-context examples (from past domains) during both fine-tuning and testing. This method is akin to Memory-replay, given its exposure to past training examples. However, its performance (when combined with DST-EGQA as DST-EGQA + IC-DST-ret) doesn't measure up to Memory-replay (with a 50 turn-level example memory size). Despite this, the third pillar demonstrates that Memory techniques and In-context fine-tuning can collectively enhance the performance.

3. **Dialogue-level Memory-replay**: Sampling examples at the dialogue level proves superior to turn-level sampling. This is attributed to the richer dialogue state a dialogue offers compared to an individual turn.

While the original paper has moments of ambiguity in presenting its main contributions and experiments, the authors managed to clarify these areas during the rebuttal phase, providing reviewers and AC with a clearer understanding of their innovations. Nonetheless, there are instances of overstatements in the paper, and I recommend the authors reevaluate and modify their claims and experimental presentation.

Regarding the scores provided: Soundness scores are uniformly at (3, 3, 3), while excitement scores varied at (4, 4, 3). It's evident that all reviewers are enthused by the proposed solution due to its evident performance improvements in the experimental settings. Additionally, the simplicity of the proposed solutions is commendable.

However, reviewers expressed concerns about:

1. The single testing benchmark.
2. Limiting their experimentation to a single model size (60M).
3. A potentially skewed comparison with the Memory-replay method.

AC's feedback on these issues:

1. **Single Testing Benchmark**: Though multiple benchmarks typically provide a more comprehensive demonstration of a solution's generalizability, the SGD is arguably the most fitting benchmark for the proposed solution. This is because other dialogue benchmarks combine domains, which may not serve as the ideal "proof-of-concept" benchmark for this study. Authors might consider other benchmarks (like a sequence of mixed-domain tasks) to further fortify their experimental claims.

2. **Model Size**: In their rebuttal, the authors mention testing larger models and observing similar findings to the 60M size. But without presenting these specific results, this concern remains unresolved.

3. **Comparison with Memory-replay**: A reviewer pointed out a potential unfair in comparison with the Memory-replay method. However, after examining lines 333-335 of the paper, it appears the authors incorporated all past examples when using the Memory-replay method. Authors might want to clarify this for enhanced transparency.

In conclusion, this study comes across as a "proof-of-concept" endeavor, mostly justifying its contributions. I consider the work to be of **moderate sound and exciting**.